# Framework for feature selection of predicting the diagnosis and prognosis of necrotizing enterocolitis

**Jianfei Song[1]☯, Zhenyu Li[2]☯, Guijin Yao[1], Songping Wei[1], Ling Li☉[1]\*, Hui Wu[2]\***

**1** College of Communication Engineering, Jilin University, Changchun, Jilin, PR China, **2** Department of Neonatology, Jilin University First Hospital, Changchun, Jilin, PR China

☯ These authors contributed equally to this work.
\* liling2002@jlu.edu.cn (LL); wuhui@jlu.edu.cn (HW)

**Data Availability Statement:** All relevant data are within the paper and its Supporting Information files.

## Abstract

Neonatal necrotizing enterocolitis (NEC) occurs worldwide and is a major source of neonatal morbidity and mortality. Researchers have developed many methods for predicting NEC diagnosis and prognosis. However, most people use statistical methods to select features, which may ignore the correlation between features. In addition, because they consider a small dimension of characteristics, they neglect some laboratory parameters such as white blood cell count, lymphocyte percentage, and mean platelet volume, which could be potentially influential factors affecting the diagnosis and prognosis of NEC. To address these issues, we include more perinatal, clinical, and laboratory information, including anemia—red blood cell transfusion and feeding strategies, and propose a ridge regression and Q-learning strategy based bee swarm optimization (RQBSO) metaheuristic algorithm for predicting NEC diagnosis and prognosis. Finally, a linear support vector machine (linear SVM), which specializes in classifying high-dimensional features, is used as a classifier. In the NEC diagnostic prediction experiment, the area under the receiver operating characteristic curve (AUROC) of dataset 1 (feeding intolerance + NEC) reaches 94.23%. In the NEC prognostic prediction experiment, the AUROC of dataset 2 (medical NEC + surgical NEC) reaches 91.88%. Additionally, the classification accuracy of the RQBSO algorithm on the NEC dataset is higher than the other feature selection algorithms. Thus, the proposed approach has the potential to identify predictors that contribute to the diagnosis of NEC and stratification of disease severity in a clinical setting.

## Introduction

Necrotizing enterocolitis (NEC) is one of the most devastating gastrointestinal diseases in the neonatal intensive care unit (NICU), with significant morbidity and mortality [1]. It is estimated that the incidence of NEC has been maintained at 3%-15%, and the mortality rate has been maintained at 20%-30% for decades [2, 3]. In general, the diagnosis of NEC is based on a combination of clinical, laboratory, and radiographic symptoms, most of which are

**Funding:** The authors received no specific funding for this work.

**Competing interests:** The authors have declared that no competing interests exist.

nonspecific or even insidious [4, 5], such as abdominal distention and reduced bowel sounds as clinical indications for feeding intolerance (FI) and NEC. These insensitive features hinder timely diagnosis and accurate treatment. Due to the difficulty of early diagnosis of NEC and the lack of reliable biomarkers, it is essential to develop an effective diagnostic model of NEC to quickly and accurately identify the key information affecting the diagnosis and prognosis of NEC, leading to more timely treatment.

NEC can be a rapidly progressing disease, and it may take only one to two days to progress from initial symptoms to full-blown illness and death. The severity of the disease is usually divided into "medical NEC" and" surgical NEC". Medical NEC refers only to medical management, while surgical NEC involves surgical intervention. In addition, as the disease progresses, the child's symptoms become more pronounced and the risk of long-term complications increases significantly, including neurocognitive impairment, developmental failure, short bowel syndrome, and cholestasis [6–8]. Therefore, it is necessary to identify high-risk infants before the disease progresses rapidly to ensure that therapeutic interventions can be initiated as soon as possible before bowel resection is required.

In recent years, machine learning (ML) methods have been widely used to diagnose cancer [9–11] and the other common diseases [12, 13]. Many researchers have developed prediction models for early NEC diagnosis (suspected NEC + NEC) and graded NEC diagnosis (medical NEC + surgical NEC). In the feature selection stage, they use statistical analysis to extract important features. In the classification stage, most researchers use ML methods such as linear discriminant analysis (LDA) [14, 15], random forest (RF) [16–18], or Light Gradient Boosting Machine (GBM) [19] as classifier models. Table 1 summarizes some studies using ML for the diagnosis or prognosis of NEC.

Most relevant studies perform well in the diagnosis and prognosis prediction of NEC. However, some key issues also need to be addressed. Firstly, most researchers use statistical methods to select features, which may ignore the correlation between features. Specifically, the idea of statistical methods is to use statistical significance to explore the association between each feature and category labels. Since there may be potential correlations between features, it is crucial to consider the correlation between features to ensure that the best performing subset of features is selected. Secondly, most researchers select a small number of features, which may overlook features that are highly correlated with predicted outcomes. Therefore, in order to solve the above problems, we need to include more features while considering their correlation.

**Table 1. Relevant studies involving ML methods for NEC diagnosis and prognosis.**

| Author | Number of features in use | classifier | AUROC |
|---|---|---|---|
| **NEC diagnosis (suspected NEC and NEC)** | | | |
| Pantalone, J. M. et al. | 14 | RF | 87.7% |
| Lure, A. C. et al. | 16 | RF | 98% |
| Jaskari, J. et al. | 14 | RF | 80.6% |
| Gao, W. J. et al. | 23 | GBM | 93.37% |
| **NEC prognosis (medical NEC and surgical NEC)** | | | |
| Ji, J. et al. | 9 | LDA | 84.38% |
| Sylvester, K. G. et al. | 27 | LDA | 81.7% |
| Pantalone, J. M. et al. | 14 | RF | 75.9% |
| Gao, W. J. et al. | 23 | GBM | 94.13% |

Abbreviations: RF, random forest; GBM, light gradient boosting machine; LDA, linear discriminant analysis.

Feature selection is a fundamental task in machine learning and statistics, and has been proved to be an effective method to process feature-related data in previous studies [20, 21]. Feature selection methods fall into three categories: filter methods, wrapper methods, and embedded methods. Filter methods [22–25] extract a subset of features from the initial dataset and use the correlation score for each feature created based on statistical measures to filter features. The advantage of this method is that the calculation is relatively easy and efficient. However, filter methods only rank features by their single-feature association with class labels and thus tend to ignore correlations between features [26]. Wrapper methods [27, 28] integrate the classification algorithm into the feature selection process. Because wrappers directly optimize the target classification algorithm, they often achieve better classification performance than filters. Wrappers usually run much slower than filter methods due to their consideration of inter-feature relationships [29]. Embedded methods [30–33] use a classification learning algorithm to evaluate the validity of features, which retain the high precision of the wrapper methods and have the high efficiency of filter methods. However, the time complexity is relatively high when processing high-dimensional data, and the redundant features cannot be completely removed [34].

To address the above issues, various works are proposed to solve feature selection problems using metaheuristics [35]. Most of them use genetic algorithms (GA) [36–39]. Meta-heuristic algorithms based on swarm intelligence are also applied to feature selection, such as ant colony optimization (ACO) [40, 41], particle swarm optimization (PSO) [42, 43], and bee swarm optimization (BSO) [44, 45]. Although metaheuristic algorithms are very effective in solving feature selection problems, the increasing number of features makes this task more and more difficult. Therefore, metaheuristic algorithms combined with machine learning and the other areas of approaches may achieve better results [46, 47].

In this paper, we propose a novel algorithm called ridge regression and Q-learning strategy based bee swarm optimization (RQBSO) metaheuristic algorithm to predict NEC diagnosis and prognosis. Ridge regression is an embedded feature selection method. Compared with the other feature selection methods, the ridge regression algorithm can filter out irrelevant features while considering the correlation between features. Therefore, the ridge regression algorithm will help to screen irrelevant variables and improve the efficiency of the meta-heuristic algorithm search. To obtain the optimal feature subset, a Q-learning strategy based bee swarm optimization (QBSO) metaheuristic algorithm is used. The advantage of Q-learning is that it does not require a complete model of the fundamental problem, because learning is performed by gathering experience referred to as trial-error [48]. By combining Q-learning with the BSO algorithm, the BSO algorithm can be adaptive in the process of searching feature subsets. In the classification stage, since the RQBSO method outputs sparse feature vectors, a linear SVM specialized in processing such data is used as the classifier model.

## Materials and methods

### Datasets

**Settings and patients.** This retrospective observational study was conducted in the neonatal intensive care unit (NICU) of Jilin University First Hospital, China, from January 1, 2015 to October 30, 2021 in accordance with the Helsinki Declaration of the World Medical Association. The study is approved by the Institutional Review Board of Jilin University First Hospital (Ethics No.2021-042). Due to the nature of the study, the informed consent from the parents/guardians of the patients is waived.

The infants with the presentation of FI who underwent abdominal imaging were enrolled, and their medical records were collected. FI is defined as "the inability to digest enteral

feedings presented as gastric residual volume of more than 50%, abdominal distension or eme-sis or both, and the disruption of the patient's feeding plan" [49]. The exclusion criteria are as follows: (a) congenital malformations, (b) spontaneous bowel perforation, (c) emergency sur-gical conditions unrelated to NEC, and (d) incomplete information.

**Data collection and definitions.** The collected NEC and FI datasets include clinical patient information obtained between diagnosis and discharge from the NICU. The final diag-nosis is determined by two independent senior neonatologists from an examination of the complete medical chart, including all perinatal and clinical findings, such as clinical manifesta-tions, laboratory tests, the results of imaging, and the disease course. In case of disagreement between the two neonatologists, a consensus is reached with the help of a senior expert. We judge whether the infant experienced NEC based on modified Bell stage ≥IIA and then deter-mine that the following criteria should be met in the whole disease course: (1) the presentation of FI; (2) abdominal signs (such as bowel sound attenuation and abdominal tenderness) and systemic signs (such as apnoea, lethargy, and temperature instability); and (3) antibiotics ther-apy and withholding feeds for at least one week [2, 50].

The NEC group is further divided into a "medical NEC group" and a "surgical NEC group". Medical NEC involves only medical management, including withholding feeds, provision of parenteral nutrition, and empirical use of antibiotics, while surgical NEC involves surgical interventions, including laparotomy and peritoneal drainage. To avoid selection bias, infants who die from severe NEC disease are assigned to the surgical NEC group. Timing of NEC onset (t0) is defined as the earliest occurrence of one of the follow-ing, within 48 hours of confirmation: 1) first notification of abdominal problems by the neonatologist, 2) abdominal radiographs or abdominal ultrasound ordered, 3) stopping enteral feeding, or 4) initiation of antibiotics [51, 52]. To identify predictors of NEC diag-nosis and disease severity, we evaluate perinatal, clinical, and laboratory variables includ-ing treatment details prior to clinical onset of NEC in detail. A detailed description of each variable is shown in the S1 Table.

## Methods

In this paper, we propose a feature selection cascade framework to address NEC diagnosis and prognosis prediction. Fig 1 shows the flowchart of our experiments, which can be divided into three stages: data preprocessing, feature selection using the RQBSO algorithm, and model classification.

All experiments are performed in a computer equipped with Jupyter notebook 3.6.1, which contains 16 GB RAM and an i7-6700 CPU clocked at 3.40 GHZ. All analyses are performed using the Scikit-learn library for Python 3.7 and the Matplotlib visualization tool.

**Data preprocessing.** First, we count the missing data and exclude clinical parameters from the study if they are missing more than 30%. Then the remaining missing values are filled using the k-nearest neighbor method. K-nearest neighbor filling is based on the principle that missing values are estimated and filled by the eigenvalues of the k nearest neighboring samples. Assuming that $x_{ai}$ (the i-th feature of the a-th sample) is a missing value, the samples that do not contain the missing value at the corresponding position will serve as providers of training information (neighbors). The reciprocal of the Euclidean distance between the a-th sample and the b-th sample is used as the weight in filling (Eq (1)).

$$\omega_{ab} = {}^{1}\!\!\left/\sqrt{\sum_k (x_{ak}-x_{bk})^2}\right.$$

(1)

where k denotes the k-th feature of the sample. The estimates of missing values can be filled

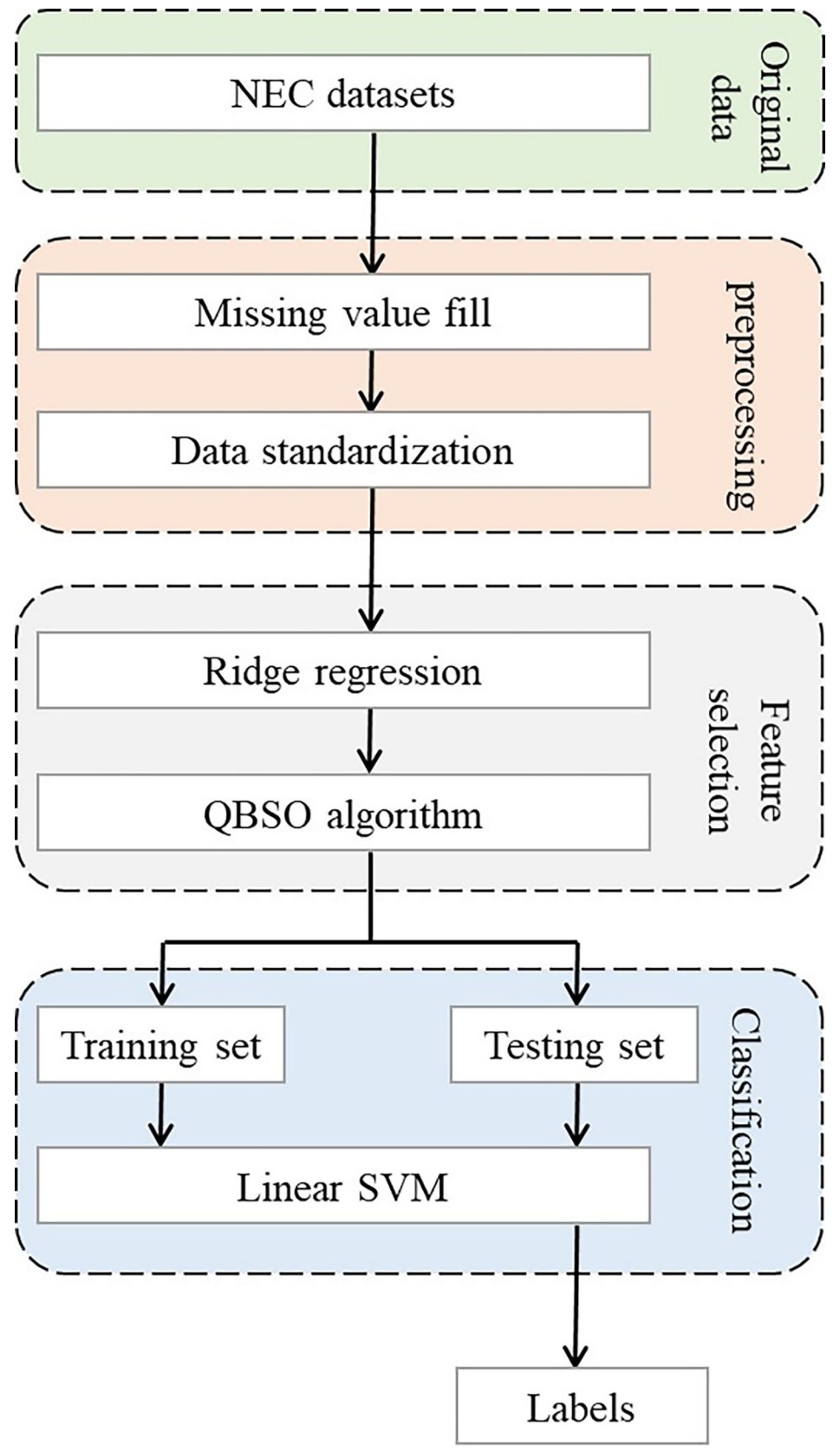

**Fig 1. The flowchart of the proposed method.**

using a weighted averaging eigenvalue of the nearest neighbor samples (Eq (2)).

$$x_{ai} = \frac{1}{\sum_b \omega_{ab}} \left( \sum_b x_{bi}\omega_{ab} \right) \tag{2}$$

Obviously, the closer the sample is to the target structure, the smaller the Euclidean distance between the two, the larger the weight factor and the greater the contribution provided to the missing value padding. However, this method has a problem that it can only estimate continuous variables, but not discrete variables. To address this drawback, we extend the existing k-nearest neighbor algorithm for estimating discrete variables by voting based on the k-nearest neighbor samples and using the nearest neighbor sample category with the most votes to fill in the missing values, as shown in Eq (3).

$$x_{ai} = Mode\{x_{bi}\}, \ b \in K \tag{3}$$

where $K$ is the set of all k nearest neighbor samples.

By adopting a hybrid strategy missing value filling method, we not only make effective use of the existing information, but also extend the application of the k-nearest neighbor feature filling method so that it can be used to fill both discrete and continuous variables.

We normalize the raw NEC data by the z-score algorithm [53] to eliminate the effects of inter-feature variation in magnitude and distribution. In addition, the normalized data can improve the convergence speed and prediction accuracy of the ML model.

**Feature selection using RQBSO algorithm.** RQBSO framework is a feature selection algorithm for the diagnosis and prognosis of NEC. It combines a ridge regression algorithm and Q-learning strategy based BSO metaheuristic algorithm. Unlike BSO, it can filter out irrelevant features by ridge regression technique, so there is no need to traverse all features in the search process of the BSO algorithm. Therefore, compared with BSO, the RQBSO algorithm has a faster training speed. Figs 2 and 3 show the structure and pseudocode of the RQBSO algorithm, respectively.

In the first stage of RQBSO, we collapse these NEC data vectors in the data input layer into a NEC data matrix suitable for processing by the feature selection algorithm. Eq (4) shows the process as

$$X = \begin{bmatrix} x_{11} & x_{12} & \cdots & x_{1N} \\ x_{21} & x_{22} & \cdots & x_{2N} \\ \vdots & \vdots & \ddots & \vdots \\ x_{m1} & x_{m2} & \cdots & x_{mN} \end{bmatrix} \tag{4}$$

where $x_{ij}$ denotes the j-th feature of the i-th sample. The matrix $X$ is fed to the next stage for features prescreening.

In the second stage of RQBSO, ridge regression is used for preliminary screening of features. The purpose is to filter out irrelevant features, reduce the space of the feature search, and improve search efficiency. The optimization objective of Ridge is

$$J(\vec{\beta}) = \sum_{i=1}^{m} (y_i - \vec{\beta}^T \cdot \vec{x_i})^2 + \lambda ||\vec{\beta}||_2^2, \ \lambda > 0 \tag{5}$$

where $\vec{x_i}$ denotes the i-th sample, $y_i$ denotes the i-th label, and the regularization parameter $\lambda$ determines the compression degree of model coefficients. We use cross-validation to determine the appropriate $\lambda$ value. To solve for the regression coefficient $\vec{\beta}$, we take the partial

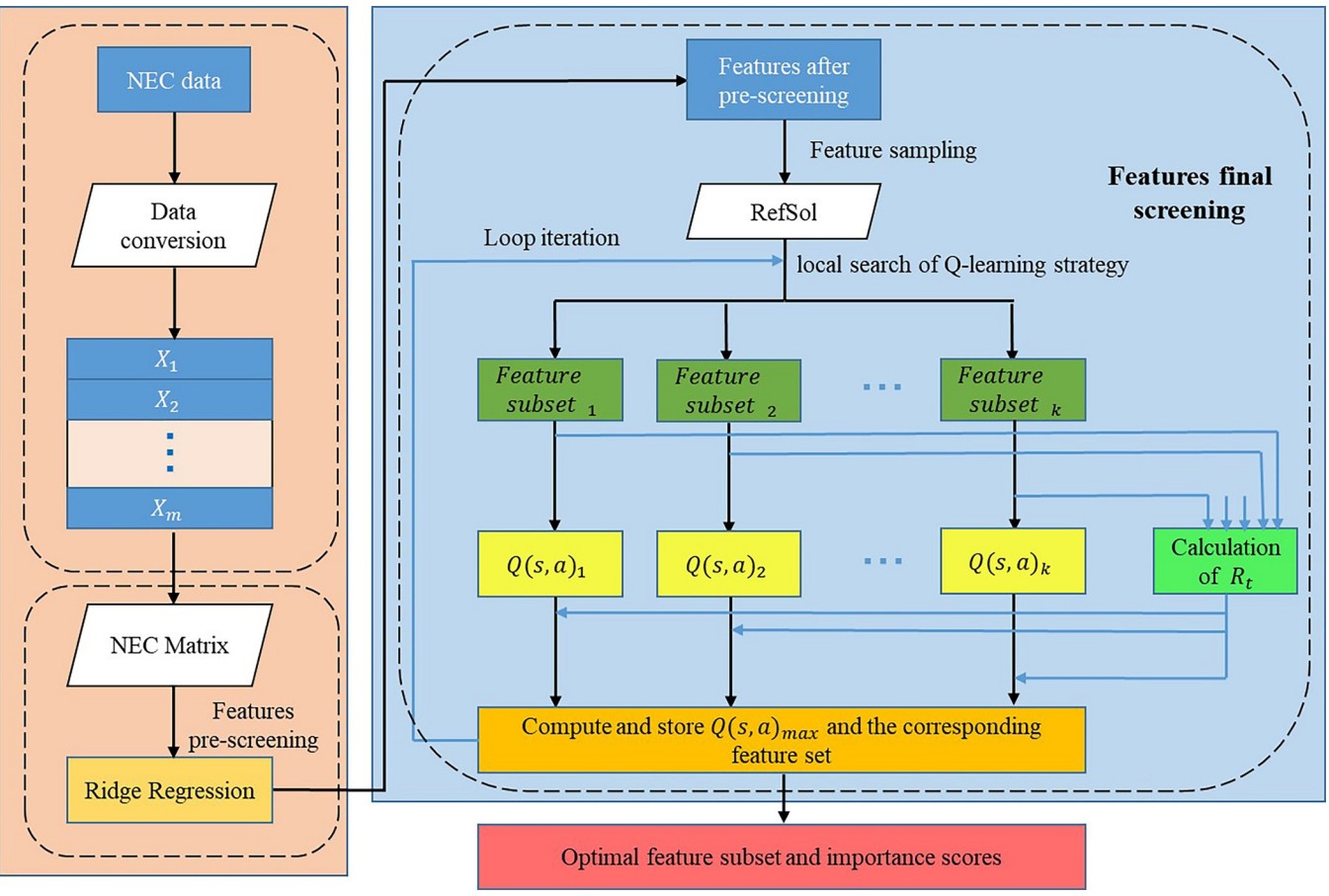

**Fig 2. The structure of the used RQBSO algorithm.**

derivative of $\overrightarrow{\beta}$ with respect to Eq (5), as shown in Eq (6)

$$J'(\overrightarrow{\beta}) = 2X^T(Y - X\overrightarrow{\beta}) - 2\lambda\overrightarrow{\beta} \tag{6}$$

where $X = [\overrightarrow{x_1}, \overrightarrow{x_2}, \ldots, \overrightarrow{x_m}]^T$, $Y = [\overrightarrow{y_1}, \overrightarrow{y_2}, \ldots, \overrightarrow{y_m}]^T$. Let $J'(\overrightarrow{\beta}) = 0$, the value of $\overrightarrow{\beta}$ can be obtained (as shown in Eq (7)):

$$\overrightarrow{\beta} = (X^TX + \lambda I)^{-1}X^TY \tag{7}$$

where $I$ denotes identity matrix. The explainable model is obtained by filtering out the features with regression coefficients equal to zero, and the purpose of feature screening is achieved.

In the final stage of RQBSO, the features that flow into the next stage are further filtered using the QBSO feature selection method to obtain the optimal subset of features. The QBSO method can be roughly divided into three stages: the determination of the search area, the local search of the Q-learning strategy, and the determination of the optimal feature subset.

*The determination of the search area.* In the first iteration, 20% of features are randomly generated as the initial feature set, which is used as the initial reference solution Refsol. To obtain the feature subset of the search area, we use two different strategies to ensure that the feature subset obtained is as different as possible. In the first strategy, the k-th feature subset is generated by flipping starting from the k-th bit of RefSol with a flipping interval of n/flip bits.

---

**algorithm 1** RQBSO

---

**Input:**
  -$X$: total NEC data(the quantity is 447);
  -$Y$: NEC diagnosis results(dataset1: 0 means FI group, 1 means NEC group; dataset2: 0 means medical NEC group, 1 means surgical NEC group)
**Output:** $F$: features finally selected by the RQBSO algorithm
  $(I)$  Data conversion:
      Form NEC Matrix $X$
  $(II)$  Ridge regression performs a preliminary feature selection on the original N features:
      $ridge\{f_1, f_2, ...f_N\} := \{f_1, f_2, ...f_n\}\,(n < N)$
  (III) QBSO performs the final screening of features:
   Generate the initial reference solution RefSol randomly
   **while** not condition of stop **do**
       Determine SearchRegion from RefSol
       Assign solutions from SearchRegion to each Bee
       **for** each Bee k **do**
           Perform a Q-local search:
           **for** each state and action **do**
               Initialize the $Q$ value table $\hat{Q}(s, a)$
           **end for**
           View current state $s$
           Loop execution
           Select action $a$ and execute $s$
           The agent receives an instant reward $r$
           View a new state $s'$
           Update the $Q$ value table $\hat{Q}(s, a)$ according to the following formula:
           $\hat{Q}(s, a) \leftarrow r + \gamma\, max\, \hat{Q}(s', a')$
       **end for**
       Choose the new reference solution following
   **end while**
   **return** $The\ Best\ feature\ subset$

---

Fig 3. **The pseudocode of the used RQBSO algorithm.**

Here, flip is a hyper-parameter, and the size of this set is equal to the number of bees, which determines the number of features to filter from Refsol. As an example, let n = 20 and filp = 5, where n denotes the number of features in Refsol. If the index of features are 0 to 19, feature subsets $f_0, f_1, f_2, f_3$ and $f_4$ are obtained by flipping the following bits, as shown in Fig 4: (0,5,10,15), (1,6,11,16), (2,7,12,17), (3,8,13,18) and (4,9,14,19). In the second strategy, the k-th subset of features is obtained by flipping n/flip contiguous bits starting from the k-th bit. Following the previous example, the feature subsets $f_0, f_1, f_2, f_3$ and $f_4$ are obtained by flipping the following bits: (0,1,2,3), (4,5,6,7), (8,9,10,11), (12,13,14,15) and (16,17,18,19). With the above two searching strategies, we determine the search area for each bee.

*The local search of the Q-learning strategy.* After determining the search area of the feature sets, we perform a nearest neighbor search for each bee by flipping each bit of the feature set separately. We denote the action of flipping the current feature as $a_t(a_t \in A_t)$ and the state as

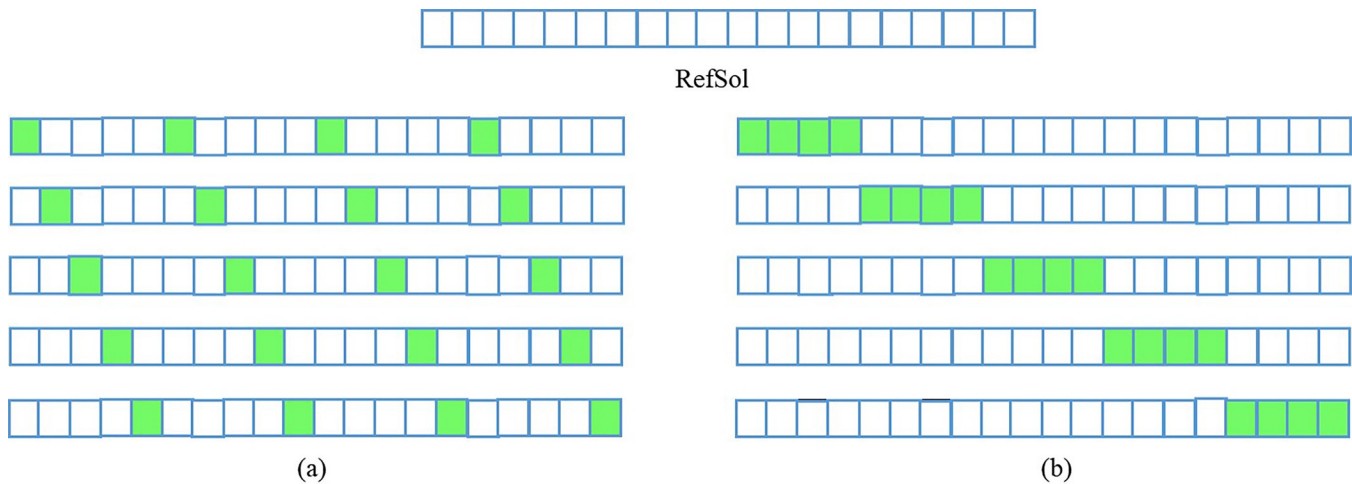

**Fig 4.** (a) solutions generated by the first strategy, (b) solutions generated by the second strategy.

$s_t(s_t \in S_t)$, where $A_t = \{a_t, a_{t+1}, \ldots a_{t+n}\}$, $S_t = \{s_t, s_{t+1}, \ldots s_{t+n}\}$, and denote $a_{t+1}$ as the action at the next moment (flipping the next feature) and $s_{t+1}$ as the state resulting from that action. By comparing the state at that moment with the next moment, we can obtain the reward $r_t$ when searching for a subset of features in different neighborhoods, as shown in Eq (8).

$$\begin{cases} r_t = Acc(s_{t+1}), Acc(s_t) < Acc(s_{t+1}) \\ r_t = Acc(s_{t+1}) - Acc(s_t), Acc(s_t) > Acc(s_{t+1}) \end{cases} \qquad (8)$$

where $Acc(s_t)$ denotes the classification accuracy of the selected feature subset in the current state, and $Acc(s_{t+1})$ denotes the classification accuracy of the selected feature subset in the next state. If the accuracy of selecting a subset of features in the current state is equal to that in the next state, then the reward $r_t$ is calculated by comparing the number of selected features in the two states, as shown in Eq (9).

$$\begin{cases} r_t = \dfrac{1}{2} * Acc(s_{t+1}), nbFeatures(s_t) > nbFeatures(s_{t+1}) \\ r_t = -\dfrac{1}{2} * Acc(s_{t+1}), nbFeatures(s_t) < nbFeatures(s_{t+1}) \end{cases} \qquad (9)$$

where $nbFeatures(s_t)$ denotes the number of features selected in the current state and $nbFeatures(s_{t+1})$ denotes the number of features selected in the next state. Then, we construct a Q-table of states and actions to store the Q-values and obtain the most favorable action (subset of features) based on the Q-values. The Q-values are calculated as shown in Eq (10).

$$Q(s_{t+1}, a_{t+1}) = r_t + \gamma * Q(s_t, a_t) \qquad (10)$$

where $0 \leq \gamma \leq 1$ denotes the discount parameter, $Q(s_{t+1}, a_{t+1})$ denotes the Q value under the next state and action, $Q(s_t, a_t)$ denotes the Q value under the current state and action, and the initial value of Q value is zero. By comparing the Q values under each feature subset, we select the feature subset with the largest Q value as the initial solution for each bee's next search, and continuously update the feature subset. The initial solution until a predetermined number of iterations is reached (localInteration), and finally return an optimal solution as the result of that bee's search.

*The determination of the optimal feature subset*. After determining the optimal solution for each bee's search, we compare its Q-value and return the feature subset corresponding to the largest Q-value as the reference solution for the next iteration. Then the (1) and (2) are repeated until a predefined number of iterations (MaxInteration) is reached. Finally, the feature set with the maximum Q-value is returned as the optimal feature subset. If the maximum Q value determined in this iteration is less than the maximum Q value of the previous iteration, we perform the diversification operation in the next iteration, that is, re-select 20% of the features at random as the solution for the next iteration, and then perform the (1) and (2) processes to determine the maximum Q value and continue the comparison with the current maximum Q value.

The advantage of the QBSO algorithm is that it processes learning through interactions with the environment. At the same time, the Q-learning adaptive searching method is used to avoid the problem of falling into local optimality.

**Model classification.** To evaluate the performance of the feature selection algorithm, we use a supervised classification model called linear SVM to calculate the classification accuracy. The linear SVM classifier is a popular supervised learning algorithm. It uses the computed decision hyperplane to classify samples. The choice of the error penalty factor, which represents the error tolerance, significantly affects the accuracy of the linear SVM. In our experiments, we use an SVM with a linear kernel function [54] and the parameter C set to 1.

## Performance measurements

To obtain a highly robust model, we use ten-fold cross-validation in our experiments. Specifically, we randomly divide the experimental data into 10 equal parts. In each experiment, 9 copies of the data are selected in turn for training and the remaining data are tested. We take the average of the 10 results as an estimate of the model accuracy.

A binary classification algorithm optimizes the parameters of a model and predicts that a new sample belongs to the positive (P) or negative (N) group. The sizes of the positive and negative groups are respectively denoted as P and N. A positive sample is defined as a true positive or false negative if it is predicted as positive or negative. A negative sample is defined as a false positive or a true negative if its prediction is positive or negative. The numbers of true positives, false negatives, false positives, and true negatives are denoted as TP, FN, FP, and TN, respectively. The binary classification performance is evaluated by the following measurements, as [55]. This study defines recall (*Rec*) as the percentages of correctly predicted positive samples, i.e. $Rec = TP/(TP+FN)$. The overall accuracy is defined as $Acc = (TP+TN)/(TP+FN+TN+FP)$. F1-score is also known as F-measure or F-score and has been widely used to evaluate the performance of a binary classification model [56]. F1-score is defined as $2*(Precision*Rec)/(Precision+Rec)$, and precision is defined as $Pre = TP/(TP+FP)$. In addition, ROC and PRC curves reflect the relationship between true positive rates and false positive rates, precision and recall, respectively. They are often used as performance graphing methods in medical decision-making [57].

## Results

### Study on the NEC cohort

Two datasets are created for analysis: dataset 1 include 447 patients, 296 (66.22%) are positive for NEC (median gestational age 31.71 (30.00–34.00) [IQR] weeks), and 151 (33.78%) are classified as FI (median gestational age 31.71 (30.14–33.85) [IQR] weeks); dataset 2 include only the NEC group (n = 296), in which a total of 91 patients (median gestational age 31.00 weeks (28.86–33.71) [IQR]) undergo surgery and 205 patients (median gestational age 32.00 weeks

(30.50–34.29)) undergo conservative treatment. Each dataset is consisted of 119 variables, and the demographic factors, clinical characteristics, and laboratory results of each dataset are shown in Table 2.

## Comparison with other feature selection algorithms

We evaluate our proposed feature selection algorithm RQBSO and compare it with three major groups of feature selection methods, including two filter methods, namely Max-Relevance and Min-Redundancy (mRMR) [58] and ReliefF [59]. Three wrapper methods, namely GA [39], BSO [44], and recursive feature elimination (RFE) [60]. Our method is also compared with two leading embedded methods, namely LASSO [30], and Ridge regression [61]. The important parameter settings of the RQBSO algorithm are shown in Table 3. The hyper-parameters of other methods are detailed in S2 Table.

Fig 5A–5D and Table 4 show the comparison of the RQBSO algorithm with three sets of feature selection methods using ten-fold cross-validation. As shown in Fig 5A and 5B, RQBSO (orange curve) outperforms the other algorithms with AUROC values of 94.20% and 91.85% on both datasets. For the same FPR level, both our method obtains a higher TPR value, which is of great significance for the diagnosis and prognosis of NEC. The AUROC values of the two filter methods (mRMR and reliefF) perform poorly due to the failure to consider the correlation between features. The PRC curves in Fig 5C and 5D also confirms these results. RQBSO has the highest AUPRC values on both datasets with 97.42% and 84.61%, respectively.

Table 4 shows that the classification accuracy of the NEC diagnosis and prognosis datasets are 91.07% and 84.37%, respectively. The advantage of our experimental accuracy is significant. In terms of accuracy, the prediction success rate of the RQBSO method exceeded 93% by conducting experiments in both dataset 1 and dataset 2. Compared with the other feature selection algorithms, our accuracy and precision are at a high level.

## Feature importance analysis

We apply the RQBSO feature selection algorithm on dataset 1 and 2 to select the optimal feature set and calculate the final ranking of the selected features. The normalized importance scores of the selected features are presented in Tables 5 and 6.

In the differential diagnosis of NEC, placenta abnormalities, platelet distribution width (PDW) at birth are the two most important features. This is followed by type of milk, lymphocyte percentage (LY%) change, signs of peritoneal irritation, achieved full enteral feeding, and drowsiness (Table 5). Overall, perinatal features account for 7.96% of the differential diagnosis of NEC, clinical features before clinical onset account for 28.84%, clinical features at clinical onset account for 25.04%, and laboratory parameters account for 38.16%.

In the classification of NEC, anemia-RBC transfusion, signs of peritoneal irritation, acidosis, tachycardia, and white blood cell count (WBC) change are the top five most important features (Table 6). Overall, perinatal features account for 9.17% of NEC classification, clinical features before clinical onset account for 27.85%, clinical features at clinical onset account for 28.06%, and laboratory parameters account for 34.92%.

## Comparison with other ML classifiers

In addition to linear SVM, we evaluate three representative classification algorithms on the dataset. The k-nearest neighbor (KNN) algorithm is a distance-based metric. The multi-layer perceptron (MLP) algorithm is one of the most widely used neural network models, and the algorithm is a multilayer feedforward neural network. The random forest (RF) algorithm is an integrated learning algorithm consisting of multiple decision trees.

**Table 2. Main perinatal and clinical characteristics of two datasets.**

| | Dataset 1 (n = 447) | | Dataset 2 (n = 296) | |
|---|---|---|---|---|
| | FI (n = 151) | NEC (n = 296) | Medical NEC (n = 205) | Surgical NEC (n = 91) |
| **perinatal characteristics** | | | | |
| GA (median [IQR], weeks) | 31.71[30.14–33.85] | 31.71[30.00–34.00] | 32.00[30.50–34.29] | 31.00[28.86–33.71] |
| BW (median [IQR], g) | 1660[1320–1920] | 1600[1100–1790] | 1660[1400–2100] | 1450[1200–1850] |
| Female (n [%]) | 48[47.1] | 59[48] | 91 [44.4] | 42[46.2] |
| BW for GA | | | | |
| SGA (n [%]) | 10[6.6] | 41[13.9] | 30[14.6] | 11[12.1] |
| AGA (n [%]) | 137[90.7] | 250[84.5] | 173[84.4] | 77[84.6] |
| LGA (n [%]) | 4[2.7] | 5[1.6] | 2[1.0] | 3[3.3] |
| Vaginal delivery (n [%]) | 72[47.7] | 127[42.9] | 81[39.5] | 43[47.3] |
| Apgar 1-min (median [IQR]) | 7[6–8] | 7[6–8] | 7[6–8] | 7[5–8] |
| Apgar 5-min (median [IQR]) | 8[8–9] | 9[8–9] | 9[8–9] | 8[7–9] |
| PPROM (n [%]) | 47[31.1] | 98[33.1] | 70[34.1] | 28[30.8] |
| Corrected GA at clinical onset (median [IQR], weeks) | 34.43[33.14–35.86] | 34.07[32.61–35.86] | 34.14[32.71–36.00] | 34.00[32.29–35.71] |
| **clinical characteristics** | | | | |
| Early Use of Antibiotics | 95[62.9] | 172[58.1] | 111[54.1] | 61[67.0] |
| MV (n [%]) | 73[48.3] | 165[55.7] | 94[45.9] | 71[78.0] |
| PDA (n [%]) | 93[61.6] | 200[67.6] | 136[66.3] | 64[70.3] |
| IVH (n [%]) | 62[41.1] | 68[23.0] | 38[18.5] | 30[33.0] |
| Infectious diseases (n [%]) | 60[39.7] | 107[36.1] | 64[31.2] | 43[47.3] |
| Anemia-RBC transfusion[a] | | | | |
| Not anemia (n [%]) | 94[63.6] | 203[68.6] | 166[81.0] | 37[40.7] |
| Anemia-not transfusion (n [%]) | 25[16.6] | 28[9.5] | 11[5.4] | 17[18.6] |
| Anemia-transfusion (n [%]) | 32[19.8] | 65[21.9] | 28[13.6] | 37[40.7] |
| *Feeding strategy* | | | | |
| Type of milk | | | | |
| human milk (n [%]) | 63[41.7] | 54[18.2] | 38[18.5] | 16[17.6] |
| Formula milk (n [%]) | 59[39.1] | 158[53.4] | 110[53.7] | 48[52.7] |
| Combination (n [%]) | 29[19.2] | 84[28.4] | 57[27.8] | 27[29.7] |
| HMF | 44[29.1] | 29[9.8] | 19[9.3] | 10[11.0] |
| Enteral nutrition start[b] | | | | |
| Quick (n [%]) | 124[82.1] | 221[74.7] | 160[78.0] | 61[67.0] |
| Medium (n [%]) | 25[16.6] | 59[19.9] | 32[15.6] | 27[29.7] |
| Slow (n [%]) | 2[1.3] | 16[5.4] | 13[6.4] | 3[3.3] |
| daily milk increment[c] | | | | |
| Quick (n [%]) | 53[35.1] | 73[24.7] | 58[28.3] | 15[16.5] |
| Slow (n [%]) | 98[64.9] | 223[75.3] | 147[71.7] | 76[83.5] |
| Probiotics | 119[78.8] | 124[41.9] | 75[36.6] | 49[53.8] |
| *Clinical manifestations* | | | | |
| Bowel sound attenuation | 60[39.7] | 182[61.5] | 121[59.0] | 61[67.0] |
| bloody stools | 81[53.6] | 105[35.5] | 75[36.6] | 30[33.0] |
| gastric residual | 39[25.8] | 141[47.6] | 97[47.3] | 44[48.4] |
| abdominal distension | 55[36.4] | 160[54.1] | 91[44.4] | 69[75.8] |
| **laboratory parameters[d]** | | | | |
| WBC at birth | 8.87[2.48–44.36] | 10.91[3.48–52.29] | 11.12[3.48–39.46] | 10.38[4.20–52.29] |
| NEUT% at birth | 0.57[0.12–0.90] | 0.58[0.06–0.93] | 0.58[0.06–0.93] | 0.57[0.15–0.84] |
| LY% at birth | 0.32[0.08–0.80] | 0.33[0.03–0.90] | 0.34[0.03–0.90] | 0.33[0.06–0.74] |

*(Continued)*

**Table 2.** (Continued)

| | Dataset 1 (n = 447) | | Dataset 2 (n = 296) | |
|---|---|---|---|---|
| | FI (n = 151) | NEC (n = 296) | Medical NEC (n = 205) | Surgical NEC (n = 91) |
| MO% at birth | 0.08[0.01–0.18] | 0.07[0–0.22] | 0.06[0.00–0.19] | 0.07[0.00–0.22] |
| NEUT# at birth | 4.78[0.5–32.4] | 5.94[0.31–43.26] | 6.04[0.31–35.00] | 5.46[1.17–43.26] |
| LY# at birth | 2.94[0.93–15.93] | 3.4[0.7–30.67] | 3.40[0.70–30.67] | 3.46[0.79–29.90] |
| MO# at birth | 0.67[0.05–5.90] | 0.67[0–4.97] | 0.62[0.00–3.40] | 0.79[0.01–4.97] |
| RBC at birth | 4.61[2.71–6.26] | 4.57[2.54–6.13] | 4.58[2.54–6.13] | 4.43[3.07–5.92] |
| HGB at birth | 172[99–237] | 172[86–226] | 173[86–226] | 170[117–220] |
| HCT at birth | 51.4[29.6–69.3] | 51.4[29–69.3] | 51.7[29.0–69.3] | 50.4[33.6–67.6] |
| MCV at birth | 111.9[98.6–129.3] | 112.9[79.2–132.9] | 112.4[97.0–209.0] | 114.4[79.2–130.6] |
| MCH at birth | 37.8[32.8–44.1] | 37.9[15.6–44.7] | 37.8[15.6–44.7] | 38.0[25.8–43.6] |
| RDW at birth | 16.45[13.9–21.1] | 16.6[13.1–26.9] | 16.7[13.1–26.9] | 16.3[13.4–25.3] |
| PLT at birth | 227[116–406] | 219[42–509] | 218[42–509] | 220[69–460] |
| PCT at birth | 0.23[0.11–0.41] | 0.23[0.09–0.55] | 0.23[0.09–0.55] | 0.24[0.10–0.46] |
| MPV at birth | 10.2[9.2–11.8] | 10.7[8.5–13.0] | 10.6[8.5–13.0] | 11.0[8.9–12.4] |
| PDW at birth | 11.1[9.3–14.6] | 11.9[8.4–18.9] | 11.8[8.4–18.9] | 12.0[8.6–15.6] |
| WBC at clinical onset | 9.71[3.44–25.37] | 9.42[0.95–48.85] | 9.72[2.07–48.85] | 8.64[0.95–27.79] |
| NEUT% at clinical onset | 0.41[0.12–0.84] | 0.61[0.14–0.91] | 0.60[0.14–0.88] | 0.62[0.18–0.91] |
| LY% at clinical onset | 0.43[0.10–0.73] | 0.27[0.06–0.73] | 0.27[0.06–0.71] | 0.26[0.07–0.73] |
| MO% at clinical onset | 0.09[0.01–0.24] | 0.08[0–0.58] | 0.08[0.00–0.58] | 0.07[0.00–0.26] |
| NEUT# at clinical onset | 3.90[0.94–16.76] | 5.61[0.39–43.02] | 5.77[0.51–43.02] | 5.02[0.39–23.50] |
| LY# at clinical onset | 3.97[0.72–8.62] | 2.47[0.06–9.53] | 2.57[0.21–7.86] | 2.25[0.06–9.53] |
| MO# at clinical onset | 0.86[0.04–3.37] | 0.68[0.01–4.43] | 0.74[0.01–4.43] | 0.54[0.05–3.69] |
| RBC at clinical onset | 3.71[2.29–5.50] | 3.86[2.41–6.08] | 3.87[2.50–6.08] | 3.74[2.41–5.03] |
| HGB at clinical onset | 125[77–180] | 135[77–310] | 136[77–310] | 126[86–185] |
| HCT at clinical onset | 37.4[23.0–51.4] | 39.6[23.7–63.0] | 40.3[23.7–63.0] | 38.4[25.3–55.4] |
| MCV at clinical onset | 101.75[83.80–113.20] | 102.65[83.60–122.80] | 103.3[85.1–122.8] | 101.2[83.6–119.4] |
| MCH at clinical onset | 34.6[28.1–39.7] | 34.9[26.7–41.0] | 35.3[26.7–41.0] | 34.0[27.0–40.6] |
| RDW at clinical onset | 15.9[13.2–20.8] | 16.01[10.30–24.30] | 16.0[10.4–24.3] | 16.3[10.3–22.4] |
| PLT at clinical onset | 317.5[105.0–823.0] | 261.5[4.0–799.0] | 257[5–609] | 272[4–799] |
| PCT at clinical onset | 0.36[0.15–0.85] | 0.32[0.01–0.91] | 0.31[0.11–0.68] | 0.33[0.01–0.91] |
| MPV at clinical onset | 11.2[9.2–13.2] | 12[9–14] | 11.96[9.50–14.00] | 12[9–14] |
| PDW at clinical onset | 13.0[8.9–20.3] | 14.2[9.6–23.0] | 14.2[9.8–23.0] | 14.5[9.6–22.8] |
| WBC change | 0.01[-0.72, 2.44] | -0.12[-0.92, 5.82] | -0.09[-0.92, 5.82] | -0.18[-0.92, 2.53] |
| NEUT% change | -0.28[-0.76, 6.00] | 0.07[-0.83, 11.81] | 0.05[-0.83, 11.81] | 0.12[-0.78, 4.06] |
| LY% change | 0.25[-0.82, 5.37] | -0.19[-0.86, 11.96] | -0.19[-0.86, 11.96] | -0.19[-0.86, 7.10] |
| MO% change | 0.17[-0.90, 17.82] | 0.25[-1.00, 800.00] | 0.33[-1.00, 800.00] | 0.18[-1.00, 500.00] |
| NEUT# change | -0.15[-0.92, 10.62] | -0.12[-0.94, 35.09] | -0.10[-0.94, 35.09] | -0.15[-0.94, 4.98] |
| LY# change | 0.25[-0.77, 6.16] | -0.27[-0.98, 3.99] | -0.24[-0.98, 2.81] | -0.35[-0.97, 3.99] |
| MO# change | 0.29[-0.95, 11.27] | 0.03[-0.99, 6400.00] | 0.14[-0.99, 6400.00] | -0.20[-0.96, 73.27] |
| RBC change | -0.18[-0.55, 0.31] | -0.13[-0.42, 0.57] | -0.13[-0.42, 0.49] | -0.15[-0.42, 0.57] |
| HGB change | -0.26[-0.63, 0.18] | -0.20[-0.55, 1.40] | -0.19[-0.48, 1.40] | -0.25[-0.55, 0.33] |
| HCT change | -0.25[-0.63, 0.15] | -0.21[-0.50, 0.36] | -0.20[-0.50, 0.36] | -0.25[-0.50, 0.30] |
| MCV change | -0.09[-0.27, -0.01] | -0.08[-0.33, 0.23] | -0.07[-0.49, 0.11] | -0.10[-0.28, 0.23] |
| MCH change | -0.07[-0.29, 0.03] | -0.06[-0.33, 1.43] | -0.06[-0.33, 1.43] | -0.09[-0.31, 0.25] |
| RDW change | -0.03[-0.21, 0.23] | -0.04[-0.39, 0.47] | -0.04[-0.39, 0.43] | -0.01[-0.37, 0.47] |
| PLT change | 0.46[-0.62, 2.38] | 0.20[-0.97, 4.21] | 0.18[-0.97, 4.21] | 0.23[-0.97, 1.93] |
| PCT change | 0.64[-0.35, 2.55] | 0.40[-0.95, 2.33] | 0.40[-0.65, 2.33] | 0.41[-0.95, 2.00] |

(*Continued*)

**Table 2.** (Continued)

| | Dataset 1 (n = 447) | | Dataset 2 (n = 296) | |
|---|---|---|---|---|
| | FI (n = 151) | NEC (n = 296) | Medical NEC (n = 205) | Surgical NEC (n = 91) |
| MPV change | 0.09[-0.16, 0.25] | 0.09[-0.14, 0.34] | 0.09[-0.09, 0.34] | 0.09[-0.10, 0.30] |
| PDW change | 0.16[-0.18, 0.81] | 0.20[-0.25, 0.97] | -0.16[-0.45, 0.63] | -0.14[-0.41, 0.53] |

Abbreviations: BW, birth weight; FPIES, Food protein-induced enterocolitis; GA, gestational age; MV, mechanical ventilation; HMF, human milk fortifier; PPROM, Preterm premature rupture of membranes; PDA, patent ductus arteriosus; IVH, intraventricular hemorrhage; IQR, interquartile range; RBC, red blood cell.

[a]Anemia is determined based on the hemoglobin concentration, the days after birth, the respiratory status and clinical manifestations based on the recommendations of Canadian Pediatric Society; The usual volume of transfusion was 10 to 20 ml kg−1 over 3 to 5 h and feeding volumes are routinely decreased during transfusions.

[b]Slow, never start or start later than postnatal day 4; Medium, start on postnatal day 3 or 4; Quick, start within postnatal day 2.

[c]Slow, the daily milk increment is less than 20 ml per kilogram of body weight until reaching full feeding volumes; quick, more than 20 ml per kilogram of body weight.

[d]laboratory values change is percentage change of each indicator at clinical onset compared with those at birth.

We use four classifiers to classify the NEC dataset. Compared with KNN, MLP and RF methods, the linear SVM has faster training and classification speed because the linear SVM is a linear classifier well suited for high-dimensional features, and it also has good generalization ability. As shown in Fig 6, the linear SVM has AUROC values of 94.22% and 91.85% and AUPRC values of 97.43% and 85.36% in datasets 1 and 2. In contrast, the AUROC and AUPRC values of KNN, MLP and RF are lower than those of the linear SVM. Therefore, the linear SVM has a significant advantage over the other three classifiers in our experiments.

## Discussion

### Predictive features

This study builds and tests a feature selection and classification algorithm that uses available data prior to disease onset for automatic diagnostic classification and NEC risk prediction. Using different ML-based classifiers trained and tested on different datasets, we obtain two general models with high accuracy and precision. In our multivariate feature selection algorithm, the previously described NEC parameters of higher WBC, signs of peritoneal irritation, and early clinical onset of NEC are significant weighted predictors of surgical NEC. Higher neutrophil percentage (NEUT%) at clinical onset, breast milk, and the use of probiotics are significant weighted predictors to identify classic NEC [7, 14, 62]. In addition, we also identify mean corpuscular hemoglobin (MCH) at clinical onset and anemia-RBC transfusion, which are risk factors for the development of NEC [63–65], as the weighted predictors of surgical NEC. This suggests that our feature selection method identifies pathophysiologically important predictors of NEC diagnosis and prognosis. Previously unreported key variables predicting NEC, such as some parameters in routine blood tests and their variations, should be brought to the attention of clinicians.

**Table 3. Hyper-parameters used by RQBSO algorithm.**

| | Parameter | value |
|---|---|---|
| Ridge | alphas | 503.15 |
| BSO | flip | 5 |
| | nBees | 10 |
| | maxIteration | 10 |
| | localIteration | 10 |
| Q-Learning | γ | 0.1 |

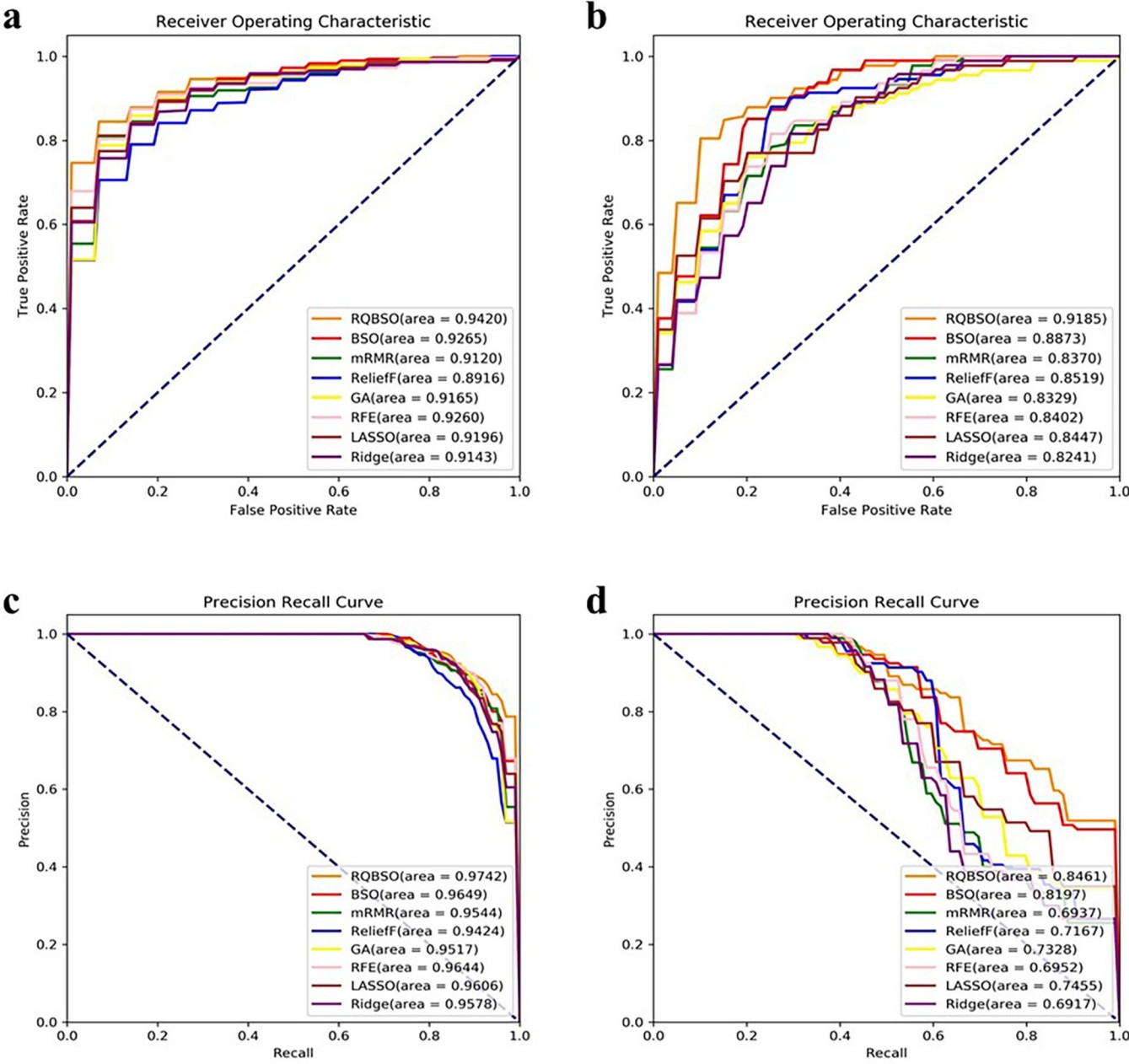

**Fig 5. Comparison of ROC and PRC curve of RQBSO and other algorithms.** (a, b) correspond to the ROC curve of dataset 1 and dataset 2. The numbers in parentheses indicate the AUROC value. The x-axis represents sensitivity, or true positive rate (TPR). The y-axis is 1-Specificity, or false positive rate (FPR). (c, d) represents the PRC curve of dataset 1 and dataset 2. The numbers in parentheses indicate the AUPRC value. The x-axis represents recall. The y-axis is precision.

## Strengths and limitations

One of the strengths of this study is the extensive collection of perinatal, clinical, and laboratory information, including topical issues in NEC in recent years, such as anemia-RBC transfusion and feeding strategies, which allows for a detailed assessment to predict the diagnosis and severity of NEC. In addition, we propose the RQBSO feature selection algorithm, which uses an integrated learning strategy that combines machine learning with a swarm optimization

**Table 4. The performance comparison of different feature selection models.**

| | RQBSO | mRMR | ReliefF | GA | BSO | RFE | LASSO | Ridge |
|---|---|---|---|---|---|---|---|---|
| **Dataset 1** | | | | | | | | |
| Acc (%) | **91.07** | 82.88 | 82.87 | 85.43 | 85.72 | 86.53 | 85.67 | 84.57 |
| Rec (%) | **96.94** | 86.26 | 88.02 | 89.86 | 94.27 | 89.82 | 89.39 | 89.39 |
| Pre (%) | **94.31** | 86.76 | 85.48 | 87.55 | 87.46 | 89.11 | 88.22 | 86.80 |
| F1-Score (%) | **92.90** | 86.36 | 86.69 | 88.57 | 89.03 | 89.41 | 88.75 | 88.00 |
| **Dataset 2** | | | | | | | | |
| Acc (%) | **84.37** | 75.40 | 76.76 | 75.53 | 81.36 | 75.02 | 77.19 | 73.37 |
| Rec (%) | 68.93 | 47.68 | 50.89 | 39.82 | **70.18** | 43.57 | 49.64 | 37.50 |
| Pre (%) | **93.33** | 70.95 | 68.71 | 70.05 | 89.31 | 65.12 | 72.25 | 65.82 |
| F1-Score (%) | **72.37** | 53.97 | 56.19 | 45.66 | 63.05 | 50.31 | 55.59 | 43.82 |

**Table 5. Feature importance ranking of dataset 1.**

| Rank | Feature | Importance score |
|---|---|---|
| 1 | Placenta abnormalities | 0.041254 |
| 2 | PDW at birth | 0.041254 |
| 3 | Type of milk | 0.040842 |
| 4 | LY% change | 0.040842 |
| 5 | Signs of peritoneal irritation | 0.040429 |
| 6 | Feeding volume at NEC onset | 0.039604 |
| 7 | Drowsiness | 0.039191 |
| 8 | NEUT% at clinical onset | 0.039191 |
| 9 | Meconium amniotic fluid | 0.038366 |
| 10 | Probiotics | 0.037954 |
| 11 | Early onset sepsis | 0.036716 |
| 12 | Acidosis | 0.036716 |
| 13 | HCT at clinical onset | 0.036304 |
| 14 | PDA | 0.035891 |
| 15 | Daily milk increment | 0.034653 |
| 16 | WBC change | 0.034653 |
| 17 | Gastric residual | 0.034241 |
| 18 | PS | 0.031766 |
| 19 | Inotropic | 0.030941 |
| 20 | Abdominal distension | 0.030528 |
| 21 | LY# change | 0.030116 |
| 22 | LY# at clinical onset | 0.029703 |
| 23 | MO# at birth | 0.028878 |
| 24 | MO% at birth | 0.028053 |
| 25 | DIC | 0.027640 |
| 26 | MCH at clinical onset | 0.025578 |
| 27 | NEUT# change | 0.025165 |
| 28 | LY% at birth | 0.021865 |
| 29 | Temperature instability | 0.021040 |
| 30 | Bloody stools | 0.020627 |

**Table 6. Feature importance ranking of dataset 2.**

| Rank | Feature | Importance score |
|---|---|---|
| 1 | Anemia-RBC transfusion | 0.069979 |
| 2 | Signs of peritoneal irritation | 0.069979 |
| 3 | Acidosis | 0.069279 |
| 4 | Tachycardia | 0.068579 |
| 5 | WBC change | 0.068579 |
| 6 | LY% at birth | 0.066480 |
| 7 | WBC at clinical onset | 0.065780 |
| 8 | Early onset sepsis | 0.061582 |
| 9 | Apgar 5-min | 0.059482 |
| 10 | PICC | 0.052484 |
| 11 | Total number of RBC transfusions | 0.049685 |
| 12 | MCH at clinical onset | 0.049685 |
| 13 | Postnatal age at clinical onset | 0.047586 |
| 14 | PLT change | 0.045486 |
| 15 | Caffeine | 0.044787 |
| 16 | Para | 0.032190 |
| 17 | NEUT# at clinical onset | 0.029391 |
| 18 | Fever | 0.025192 |
| 19 | MCV at clinical onset | 0.023793 |

algorithm. This algorithm achieves better feature selection results on both the NEC diagnosis and risk prediction datasets. The average classification accuracy of RQBSO filtered features is higher in both datasets. Moreover, most of the features filtered by RQBSO are clinically significant, and these important weighted predictive values deserve the attention of clinicians.

The present study has some limitations. Firstly, the number of extracted features is disproportionate to the size of the dataset, which may affect the performance of our ML classifiers, and increasing the sample size would probably improve performance. Secondly, the Bell staging criteria used in this study provide a relatively poor description of bowel injury. Although we exclude possible confounding factors when separating medical NEC from surgical NEC, applying ML methods to classify datasets with poorly defined non-discrete entities may be flawed. Finally, the lack of out-of-sample validation and the single-center retrospective design make our models less applicable. We hope to validate our models with future data from our NICU or other NICUs.

## Conclusion

In this work, we propose a new feature selection framework RQBSO for early diagnosis of NEC and identification of high-risk infants. To evaluate the effectiveness of our algorithms, we conduct experiments on two skewed datasets of NEC differential diagnosis and risk prediction. In the end, we classify the NEC differential diagnosis data with an average recognition accuracy of 91.07% and an AUROC value of 94.20%. While the accuracy of the other set is only 84.37%, and the AUROC value is 91.85%. The experimental results show that the method has a high recognition accuracy in the differential diagnosis and risk prediction of NEC. In addition, the method screens some new significant weighted predictors that may lead to earlier identification and more timely treatment.

In future work, we plan to apply our method to higher-dimensional datasets and perform deeper parameter tuning to investigate their impact on algorithm performance.

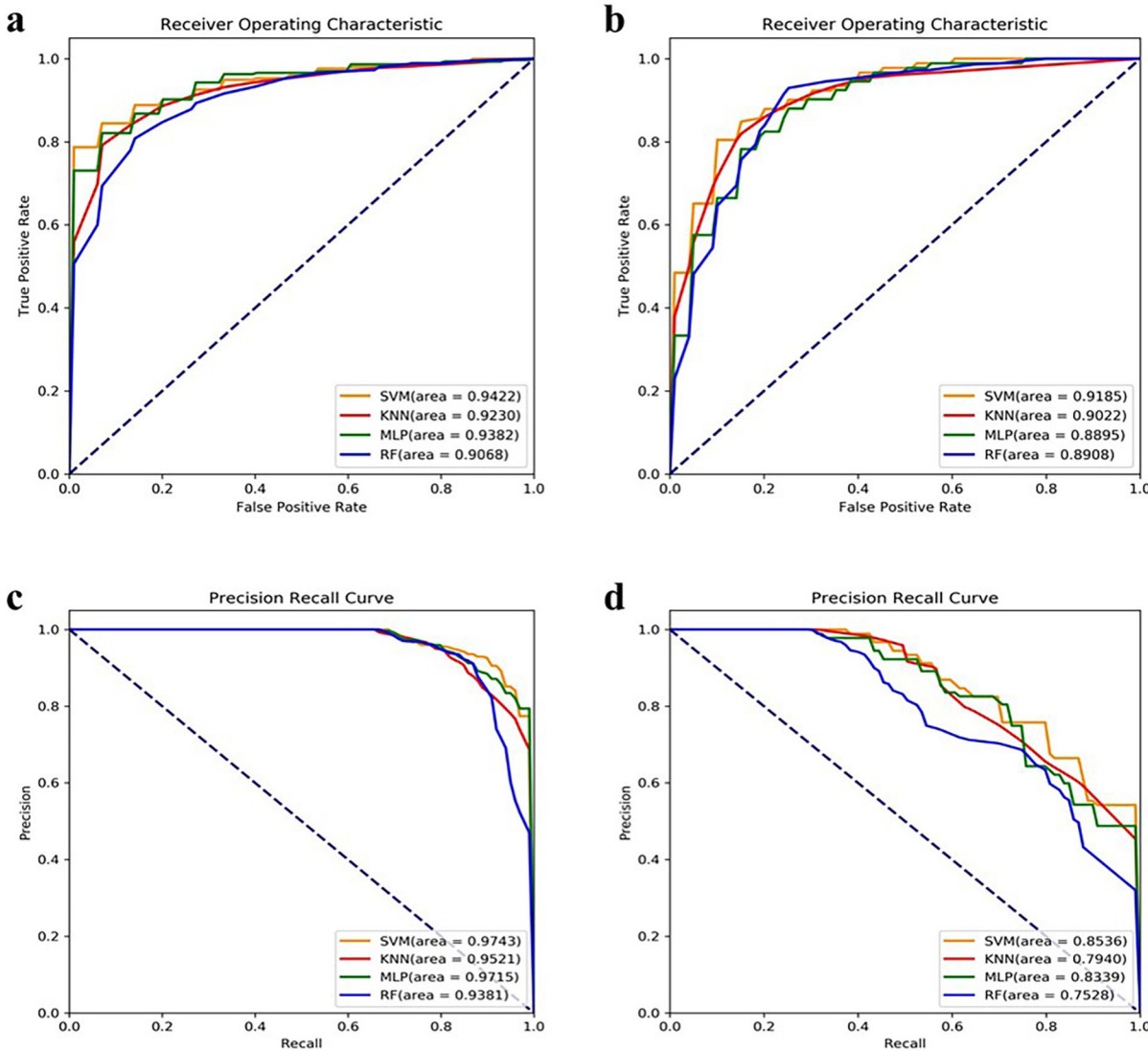

**Fig 6. Comparison of ROC and PRC curve of different classifiers.** (a, b) correspond to the ROC curve of dataset 1 and dataset 2. The numbers in parentheses indicate the AUROC value. The x-axis represents sensitivity, or true positive rate (TPR). The y-axis is 1-Specificity, or false positive rate (FPR). (c, d) represents the PRC curve of dataset 1 and dataset 2. The numbers in parentheses indicate the AUPRC value. The x-axis represents recall. The y-axis is precision.

## Supporting information

**S1 Table. Description of different features.**
(DOCX)

**S2 Table. Hyper-parameters used by other algorithms.**
(DOCX)

**S1 Dataset.**
(ZIP)

## Acknowledgments

We would like to thank colleagues at the Neonatology Department of the First Hospital of Jilin University for robust data support.

## Author Contributions

**Conceptualization:** Jianfei Song, Zhenyu Li, Guijin Yao, Ling Li, Hui Wu.

**Data curation:** Zhenyu Li, Songping Wei, Hui Wu.

**Formal analysis:** Jianfei Song, Songping Wei.

**Investigation:** Hui Wu.

**Methodology:** Jianfei Song, Zhenyu Li, Guijin Yao, Ling Li.

**Software:** Jianfei Song, Songping Wei.

**Supervision:** Guijin Yao, Songping Wei, Ling Li.

**Validation:** Jianfei Song, Ling Li.

**Visualization:** Jianfei Song.

**Writing – original draft:** Jianfei Song, Zhenyu Li.

**Writing – review & editing:** Guijin Yao, Ling Li, Hui Wu.

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
