## [Decision Letter · Decision Letter 0]

6 Jun 2022

PONE-D-22-08369Predicting the diagnosis and prognosis of necrotizing enterocolitis using a novel feature selection frameworkPLOS ONE

Dear Dr. Li,

Thank you for submitting your manuscript to PLOS ONE. After careful consideration, we feel that it has merit but does not fully meet PLOS ONE’s publication criteria as it currently stands. Therefore, we invite you to submit a revised version of the manuscript that addresses the points raised during the review process.Please submit your revised manuscript by Jul 21 2022 11:59PM. If you will need more time than this to complete your revisions, please reply to this message or contact the journal office at plosone@plos.org. Please include the following items when submitting your revised manuscript:A rebuttal letter that responds to each point raised by the academic editor and reviewer(s). You should upload this letter as a separate file labeled 'Response to Reviewers'.A marked-up copy of your manuscript that highlights changes made to the original version. You should upload this as a separate file labeled 'Revised Manuscript with Track Changes'.An unmarked version of your revised paper without tracked changes. You should upload this as a separate file labeled 'Manuscript'.

We look forward to receiving your revised manuscript.

Kind regards,

Vijayalakshmi Kakulapati, Ph.D

Academic Editor

PLOS ONE

Journal Requirements:

Reviewers' comments:

Reviewer's Responses to Questions

**Comments to the Author**

1. Is the manuscript technically sound, and do the data support the conclusions?

Reviewer #1: Partly

Reviewer #2: Yes

2. Has the statistical analysis been performed appropriately and rigorously? 

Reviewer #1: I Don't Know

Reviewer #2: Yes

3. Have the authors made all data underlying the findings in their manuscript fully available?

Reviewer #1: Yes

Reviewer #2: Yes

4. Is the manuscript presented in an intelligible fashion and written in standard English?

Reviewer #1: Yes

Reviewer #2: Yes

5. Review Comments to the Author

Reviewer #1: 1. Line 1: Based on the content of the work, I will rather suggest the author coin the topic in line with the modified algorithm RQBSO. Aside this, the work will expose the novelty not the topic. However, I suggest something like “Framework for feature selection of predicting the diagnosis and prognosis of nectrotizing enterocolitis”

2. Line 23-26: list some potential influencing factors ignored

3. Line 29—31: What is RQBSO? NEC? And does linear support vector machine translate to SVM?

4. Line 47: I’m not aware this type of citation style

5. Line 54: Check the sentence especially the use of the word “mine”

6. Line 74: Take abbreviation outside the table as note

7. Line 82: Second or secondly

8. Line 84: kindly justify why the need to improve prediction requires inclusion of more features

9. Line 91-94: consider review into smaller sentences

10. Line 112: This suggest the main focus of this work, the abbreviations need to be check especially how line 113-114 translate to RQBSO

11. Line 127: from this stage the heading and heading numbering are defective. Check the heading style and heading numbering of the journal. Specifically, heading line 127 and line 177 should go together and each description of information in it must flow and justified

12. Line 149: The patient characteristics should have gone with line 128, Patient and data sets

13. Line 155-156. This covers about four pages of data. These data shouldn’t have come under material used or patient characteristics rather data presentation or during analysis. If it should be here kindly justify

14. Line 178-179: the proposed method include three steps-two steps mentioned and the heading that follows are more than three. Coordinate and arrange your materials to follow the proposed steps

15. Line 182 onward contained mathematical expressions which the character and formatting impair the quality of information in the equation. For example, in line 188, the multiplication sign is not far from the other x’s define in the equation

16. Line 194: Reconcile eqt2 because there will be a problem if it is substituted into eqt1

17. Line 202: check eqt3

18. Line 211: in this work we use the RQBSO algorithm……., if RQBSO is the main focus, it must have been describe before this place

19. Line 230: delete Eq.(5)

20. Line 223-241: You need to utilize software that can improve your mathematical writings to make sense, this applies to all equations in the work

21. Line 250, 268 and 298: headings

22. Line 312: the entirty of these section should have been devoted to development and justification of RQBSO where the figure will now be pictorial representation of the proposed method. As it stand now, this section is the pillar of the work which needs to be strengthened

23. Lines 314, 319, 329, 336, 341, 347 should go to introduction or still reeiw of relevant tools but not where the main work is been discussed. It can only be mentioned to justify its use

24. Line 353-356: deliberate action is needed to cite and justify these equations

25. Line 358: Evaluation cannot be done under the heading Result and Discussion. Is either the heading or content is faulty

26. Line 357 and line 441: reconcile

27. Line 470: Here, we can simply be ……’’In this work, a novel………’’

28. Line 478: the word … most… there must be specific by mentioning the existing ……

General Comment

My adventure of going through this work suggest to me that the author has lots of materials to be presented to justify publication. Unfortunately, the materials were not well organized and presented. The language of communication is good but research writing and presentation is lack which hindere the flow of communication. Aside this

1. Style and formatting

2. Heading and heading numbering

3. Citations referencing style (I don’t know if this is the journal style)

4. Probably due to submission, I suggest all part are put together before released for reviewing

Reviewer #2: The manuscript has very well and enough introduction, all the analysis and calculations are made in good manner. The figures are in a good resolution and all the references are put in order of date. The manuscript has got accepted from my side.

6. PLOS authors have the option to publish the peer review history of their article (what does this mean?). If published, this will include your full peer review and any attached files.

Reviewer #1: **Yes: **MKA Abdulrahman

Reviewer #2: No

---

## [Author Response · Author response to Decision Letter 0]

20 Jul 2022

Dear Editor and Reviewers:

Thank you for your letter and for the reviewers’ comments concerning our manuscript entitled “Predicting the diagnosis and prognosis of necrotizing enterocolitis using a novel feature selection framework” (Manuscript Number: PONE-D-22-08369). We have studied the comments carefully, and they have all been valuable and very helpful for revising and improving our paper. We have revised the article based on the recommendations of the reviewers and the revised portions are marked in red in the paper. We hope that the revision is acceptable and look forward to hearing from you. 

Review Comments to the Author

Reviewer #1:

Comment 1: Line 1: Based on the content of the work, I will rather suggest the author coin the topic in line with the modified algorithm RQBSO. Aside this, the work will expose the novelty not the topic. However, I suggest something like “Framework for feature selection of predicting the diagnosis and prognosis of nectrotizing enterocolitis”.

Reply 1: Thank you for your comment. As you said, the focus of our work is to propose the RQBSO algorithm and use it to predict the diagnosis and prognosis of NEC, as well as to uncover some potential impact factors. At the same time, our work exposes freshness rather than theme. Therefore, using “Framework for feature selection of predicting the diagnosis and prognosis of necrotizing enterocolitis” as the title can better highlight the focus of our work. We revise it according to your suggestion.

Actual changes: 

Changing the title of Line 1.

Comment 2: Line 23-26: list some potential influencing factors ignored.

Reply 2: Thank you for your comment. Previous studies have focused on perinatal characteristics (gestational age, birth weight, etc.) and clinical characteristics (blood and stool, mechanical ventilation, etc.) of patients while neglecting some laboratory parameters such as white blood cell count, lymphocyte percentage, and mean platelet volume. In addition, our study incorporates recent topical issues such as anemia-RBC transfusion and feeding strategies, allowing for detailed assessment of possible variables predictive of the diagnosis and severity of NEC.

Actual changes:

The influencing factors ignored by previous studies are added after Line 26.

Comment 3: Line 29—31: What is RQBSO? NEC? And does linear support vector machine translate to SVM?

Reply 3: Thank you for your comment. I apologize that the explanation of the RQBSO algorithm and linear SVM in the previous manuscript raised questions for you. The RQBSO algorithm means a ridge regression and Q-learning strategy based bee swarm optimization metaheuristic algorithm. NEC is a worldwide pediatric disease and a major source of neonatal morbidity and mortality. There are some differences between linear SVM and SVM. More precisely, the SVM module is a wrapper for the libsvm library and supports different kernel functions (linear kernel, Gaussian kernel, Laplace kernel, etc.), while linear SVM is based on liblinear and supports only linear kernel functions. In the revised version, I correct the writing of RQBSO and linear SVM so that you can understand it better.

Actual changes:

Correcting the definition of RQBSO in Line 29.

Correcting the definition of linear SVM in Line 31.

The definition of NEC is described in detail in Line 21.

Comment 4: Line 47: I’m not aware this type of citation style.

Reply 4: Thank you for your comment. The formatting requirements for references in PLOS one journals are as follows:

1. References should be listed after the main text, before the supporting information.

2. References with more than six authors should list the first six author names, followed by “et al.”

3. References should be formatted according to the NLM/ICMJE style: https://www.nlm.nih.gov/bsd/uniform_requirements.html.

In the references of the previous manuscript, I used the NLM style supported by the PLOS one journal. It is possible that this style is used by fewer authors, so in the revised manuscript I change the citation style of the references to the ICMJE style supported by the PLOS one journal.

Actual changes:

Revising the format of references in Lines 491-720.

Comment 5: Line 54: Check the sentence especially the use of the word “mine”.

Reply 5: Thank you for your comment. In the previous manuscript, the original meaning of the sentence was that the diagnosis of NEC in clinical medicine currently suffers from the difficulty of early diagnosis and the lack of reliable biomarkers. Therefore, there is an urgent need to develop effective NEC diagnostic models to quickly and accurately identify relevant information affecting the diagnosis and prognosis of NEC, thus enabling a more accurate diagnosis. Therefore, the word “mine” is not used appropriately. In the revised manuscript, I replace the word “mine” and phrase the sentence in more detail.

Actual changes:

Changing the word "mine" in line 54, and phrasing the sentence in more detail.

Comment 6: Line 74: Take abbreviation outside the table as note.

Reply 6: Thank you for your suggestion. In the previous manuscript, we did not annotate the abbreviations for the classifiers in Table 1, but gave the full name of the classifier in its first occurrence in the table. That's not quite the norm. In the revised manuscript, instead of writing the full name of the classifier in the table, we use abbreviations and write the abbreviations of the classifier as comments outside the table.

Actual changes:

Changing the full name of the classifier to an abbreviation in Table 1 in Line 74, and adding a comment outside the table.

Comment 7: Line 82: Second or secondly.

Reply 7: Thank you for your comment. There is a small syntax problem here, and secondly should be used instead of second.

Actual changes:

Changing the word "second" to "secondly" in Line 82, and checking the paragraph thoroughly.

Comment 8: Line 84: kindly justify why the need to improve prediction requires inclusion of more features.

Reply 8: Thank you for your comment. After careful examination, I found some problems with the description of this sentence. Previous studies have the following problems: First, they mostly used statistical methods to select features, which may ignore the correlation between features. Secondly, most of the researchers selected a small number of features, which may ignore features that are highly correlated with predicted outcomes. Therefore, in order to fully consider the association of unknown features and to explore the potential influencing factors associated with NEC diagnosis and disease severity stratification, we need to include more features for study. It is not certain that the inclusion of more features will improve prediction. Therefore, in the revised manuscript, I will remove the statement of improved prediction.

Actual changes:

Removing the statement of improved prediction in Line 84.

Comment 9: Line 91-94: consider review into smaller sentences.

Reply 9: Thank you for your suggestion. In the previous manuscript, I used long sentences to describe the filter method and did not show the advantages and disadvantages of the method well. As a result, it may be relatively difficult to understand. In the revised manuscript, I rewrite the long sentences into shorter sentences and describe the advantages and disadvantages of the method so that the method can be better understood.

Actual changes:

Changing long sentence to short sentence in Lines 91-94 and thoroughly checking the paragraph.

Comment 10: Line 112: This suggest the main focus of this work, the abbreviations need to be check especially how line 113-114 translate to RQBSO.

Reply 10: Thank you for your comment. I apologize that the abbreviation of the RQBSO algorithm in the previous manuscript raised questions for you. In the revised version, I redefine the abbreviation of the RQBSO algorithm. The RQBSO algorithm means a ridge regression and Q-learning strategy based bee swarm optimization metaheuristic algorithm. The ridge regression algorithm allows filtering out irrelevant features while considering the correlation between features. Thus, the ridge regression algorithm will help filter out irrelevant variables and improve the efficiency of the metaheuristic algorithm search. Using the Q-learning strategy based bee swarm optimization metaheuristic algorithm, adaptive learning can be performed during the search for feature subsets to obtain the optimal feature subset.

Actual changes:

Redefining the abbreviation of the RQBSO algorithm in Line 112.

Comment 11: Line 127: from this stage the heading and heading numbering are defective. Check the heading style and heading numbering of the journal. Specifically, heading line 127 and line 177 should go together and each description of information in it must flow and justified.

Reply 11: Thank you for your suggestion. In the original draft, I wrote Materials and Methods in two separate sections. As you said, the headings and the numbering of the headings are somewhat flawed and hinder the reader's reading and understanding. In the revised version, I integrate Materials and Methods together. Specifically, in Materials, I first present the ethical instructions and patient exclusion criteria, then describe the data collection process and give a detailed description of the relevant medical terminology. In Methods, I first introduce the entire experimental procedure, then provide a detailed description of each process in the experimental procedure. Finally, I give the performance evaluation metrics used in the study.

Actual changes:

Combining the headings of Lines 127 and 177, and detailing the datasets used and the methodology in that section.

Comment 12: Line 149: The patient characteristics should have gone with line 128, Patient and data sets.

Reply 12: Thank you for your comment. Both sections Patients and datasets and Patient characteristics provide patient-specific information, including patient exclusion and inclusion criteria, descriptions of relevant medical terminology, and personal information (characteristics) of the patient, and should therefore be integrated. In the revised version, I describe Patients and datasets in more detail, including the ethical statement, the inclusion and exclusion criteria for patients, the data collection process, and the introduction of relevant medical terminology. The patient characteristics are also statistically analyzed and then shown in the data analysis.

Actual changes:

Combining the content of Line 128 and Line 149 and rewriting the datasets section.

Comment 13: Line 155-156. This covers about four pages of data. These data shouldn’t have come under material used or patient characteristics rather data presentation or during analysis. If it should be here kindly justify.

Reply 13: Thank you for your comment. You are correct. In the previous manuscript, I used statistical analysis to provide a thorough description of the patient's clinical characteristics, specifically the patient's perinatal characteristics, clinical features, and laboratory parameters. The categorical variables were summarized as counts and percentages, whereas the non-normally distributed continuous variables were summarized as medians and quartiles. Routine blood test results are analyzed as percentage change at the onset of NEC compared to birth. And this part should not belong to the Materials used but to the period of data presentation or analysis. In the revised manuscript, I put this part of the statistical analysis into the Study on the NEC cohort under Results. This section focuses on the clinical characteristics of the patients.

Actual changes:

The contents of Table 2 in rows 155-156 are placed in the "NEC Cohort Study" under the "Results" section, and the statistical analysis process is described in more detail in this section.

Comment 14: Line 178-179: the proposed method include three steps-two steps mentioned and the heading that follows are more than three. Coordinate and arrange your materials to follow the proposed steps.

Reply 14: Thank you for your comments. In the previous manuscript, I proposed a framework consisting of three steps of data pre-processing, feature selection and classification, but the method was followed by more than three subheadings (K-fold cross-validation and Evaluation indicators were added). Therefore, in the revised version, I made the following changes: k-fold cross-validation and Evaluation were integrated into a secondary heading after Datasets and Methods in order to make the reader better understand the core content of the article.

Actual changes:

The subheadings of the proposed method in Lines 178-179 are revised and the method is described in more detail.

The “K-fold cross-validation” section in Line 341 is integrated with the “Evaluation indicators” section in Line 347 and placed in the section after “Datasets” and “Methods”.

Comment 15: Line 182 onward contained mathematical expressions which the character and formatting impair the quality of information in the equation. For example, in line 188, the multiplication sign is not far from the other x’s define in the equation.

Reply 15: Thank you for your suggestion. In the previous manuscript, I was not sufficiently rigorous in writing certain mathematical expressions, resulting in characters and formatting of certain expressions that may compromise the quality of the information in the equations. Therefore, in the revised manuscript, I check all the mathematical expressions in the article and rewrite all the equations. For example, to address the problem that the multiplication sign in line 188 in the original draft was easily confused with other x's in the equation, I change the multiplication sign to "*" to better distinguish them.

Actual changes:

The mathematical expressions after Line 182 are thoroughly checked and all equations are rewritten.

Comment 16: Line 194: Reconcile eqt2 because there will be a problem if it is substituted into eqt1.

Reply 16: Thank you for your comment. Eqt1 and eqt2 are formulas for missing value filling using the K-nearest neighbor algorithm, which works by estimating the missing values from the eigenvalues of the k nearest neighbor samples and filling them. In the previous manuscript, I was not clear enough about the formulation, thus causing confusion to you. In the revised manuscript, I describe the k-nearest neighbor algorithm in more detail and adjusted eqt1 and eqt2. First, we use the reciprocal of the Euclidean distance between two samples as the padding weight (eqt1). Then we calculate the Euclidean distance between the sample with missing values and the other samples to determine the nearest neighbor sample of that sample. Finally, the weighted average of the eigenvalues of the nearest neighbor samples is used to fill in the estimates of the missing values (eqt2).

Actual changes:

The eqt1 in Line 188 and eqt2 in Line 194 are rewritten and the k-nearest neighbor algorithm is described in more detail.

Comment 17: Line 202: check eqt3

Reply 17: Thank you for your comment. Eqt3 is an extension of our existing k-nearest neighbor algorithm for estimating discrete variables. In the revised version, I modify eqt3 based on eqt1 and eqt2. First, the nearest neighbor samples of samples with missing values are calculated according to Euclidean distance. Secondly, the nearest neighbor samples are voted based on the nearest neighbor samples, and the nearest neighbor sample category with the most votes is filled with missing values.

Actual changes:

Modifying eqt1 in Line 188 and eqt2 in Line 194 and extending eqt3 based on the modified eqt1 and eqt2.

Comment 18: Line 211: in this work we use the RQBSO algorithm……., if RQBSO is the main focus, it must have been describe before this place.

Reply 18: Thank you for your comment. This section is an introduction to the RQBSO algorithm and a description of the specific process of the algorithm: the RQBSO framework is a feature selection algorithm for NEC diagnosis and prognosis. It combines a ridge regression algorithm and a BSO metaheuristic based on a Q-learning strategy. And "in this work we use the RQBSO algorithm ......." is a summary statement. It is inappropriate to be placed here. Therefore, in the revised version, I removed the relevant summary statement and described the specific process of the RQBSO algorithm in detail in this section.

Actual changes:

The summary statement in Line 211 is removed and the specific procedure of the RQBSO algorithm is described in detail in that section.

Comment 19: Line 230: delete Eq.(5).

Reply 19: Thank you for your suggestion. In the revised version, I remove Eq. (5).

Actual changes:

Removing Eq.(5) from line 230.

Comment 20: Line 223-241: You need to utilize software that can improve your mathematical writings to make sense, this applies to all equations in the work.

Reply 20: Thank you for your comment. In the revised draft, I check all the formulas and replace the software used to write the mathematical formulas so that the reader could read them better.

Actual changes:

Rechecking the formulas in Lines 223-241 and replacing the software used to write the mathematical formulas.

Comment 21: Line 250, 268 and 298: headings.

Reply 21: Thank you for your comment. Due to my lack of standardization in writing, the numbers (1), (2), and (3) in lines 250,268 and 298 of the original manuscript may have been interpreted by you as subheadings. However, these three lines are not a subheading, but the execution flow of the QBSO algorithm. In the revised version, I bold these three lines to indicate that the line is the execution process of the algorithm.

Actual changes:

Deleting (1), (2), (3) in Lines 250, 268 and 298. And bolding the three lines.

Comment 22: Line 312: the entirty of these section should have been devoted to development and justification of RQBSO where the figure will now be pictorial representation of the proposed method. As it stand now, this section is the pillar of the work which needs to be strengthened.

Reply 22: Thank you for your comment. As you said, Fig. 4 is the pseudocode of the RQBSO algorithm, which aims to describe the whole execution of the algorithm in a form close to natural language. And this part is also the core and backbone of the whole algorithm. Therefore, in the revised version, I have repositioned Fig. 4 and described the RQBSO algorithm in more detail so that the reader can better understand the execution of the algorithm.

Actual changes:

Repositioning Fig. 4 under the “Feature selection” section in Line 210, and describing the RQBSO algorithm in more detail.

Comment 23: Lines 314, 319, 329, 336, 341, 347 should go to introduction or still reeiw of relevant tools but not where the main work is been discussed. It can only be mentioned to justify its use.

Reply 23: Thank you for your comment. Lines 314, 319, 325, 329, and 336 focus on the ML models used in the model prediction phase and compare the four models mentioned above. Whereas we end up using the linear SVM model, the remaining three models are not the place to discuss the main work. Therefore, in the Model classification section of the revised manuscript, I only present the linear SVM model, while the Comparison with other ML classifiers section in Results shows the comparison of the four models. And lines 341 and 347 are about the performance evaluation metrics used in the study. In this paper, we use recall, accuracy, precision, F1 score, ROC curve, and PRC curve to evaluate the performance of model prediction. And the above metrics are also the most common evaluation metrics in binary classification performance.

Actual changes:

Removing the description of other ML models in Lines 325, 329 and 336, and keeping only the description of the linear SVM model in line 319.

The content of Lines 341 and 347 are merged and the merged content is modified.

Comment 24: Line 353-356: deliberate action is needed to cite and justify these equations.

Reply 24: Thank you for your comment. This study uses recall, accuracy, F1-score, precision, ROC curve and PRC curve to measure the prediction performance of the model. Where recall is defined as the percentage of positive samples correctly predicted. Accuracy is defined as the percentage of positive samples predicted accurately. Precision is defined as the percentage of correct predictions among all samples. F1-score is defined as the harmonic mean of the accuracy and recall rates. the ROC and PRC curves reflect the relationship between the true and false positive rates, the precision rate and the recall rate, respectively. The above metrics have been widely used to evaluate the performance of binary classification models, and the ROC and PRC curves are often used as performance mapping methods in medical decision making. In the revised manuscript, references and proofs regarding the above equations are listed with relevant literature supporting them.

Actual changes:

Giving proofs about equations in lines 353-356 with references to support them.

Comment 25: Line 358: Evaluation cannot be done under the heading Result and Discussion. Is either the heading or content is faulty.

Reply 25: Thank you for your comment. You are correct. In the previous manuscript, I evaluated the RQBSO algorithm and other feature selection algorithms under Results and discussion, which is not appropriate. I should have evaluated and compared the performance of the methods when comparing the performance of different feature selection algorithms. I have adjusted this section in the revised manuscript.

Actual changes:

Removing the evaluation of the performance of different algorithms in Line 358 and adding it to Line 380 under the heading Comparison with other feature selection algorithms.

Comment 26: Line 357 and line 441: reconcile.

Reply 26: Thank you for your comment. In the revised version, I reconcile the Results and discussion chapter. Specifically, I split this section into two specific chapters. The Results section focuses on a statistical analysis of data, comparison with different methods, and the importance of analytic features. The Discussion section focuses on some of the pathophysiologically important predictors of NEC diagnosis and prognosis that are explored, as well as the strengths and limitations of the study.

Actual changes:

Dividing the heading of Lines 357 and 441 into two sections and refactoring the content of both sections.

Comment 27: Line 470: Here, we can simply be ……’’In this work, a novel………’’.

Reply 27: Thank you for your suggestion. There are writing irregularities in this section, and I revise it according to your comments.

Actual changes:

Simplifying the content in Line 470.

Comment 28: Line 478: the word … most… there must be specific by mentioning the existing ……

Reply 28: Thank you for your comment. In this study, the RQBSO algorithm is proposed and experimented on two skewed datasets for NEC differential diagnosis and risk prediction. To evaluate the effectiveness of our algorithm, we compare it with three sets of feature selection methods and our algorithm outperforms the other three sets of algorithms. However, the use of the word "most" is not appropriate because the number of compared algorithms is limited and the existing algorithms need to be mentioned to specify them. Therefore, in the revised manuscript, I delete "which is better than most existing feature selection algorithms" and instead indicate that our method has a high recognition accuracy in differential diagnosis and risk prediction of NEC.

Actual changes:

The phrase "which is better than most existing feature selection algorithms" in Line 478 is removed, and instead, our method is shown to have a high recognition accuracy in differential diagnosis and risk prediction of NEC.

Reviewer #2: The manuscript has very well and enough introduction, all the analysis and calculations are made in good manner. The figures are in a good resolution and all the references are put in order of date. The manuscript has got accepted from my side.

Reply: Thank you very much for your recognition of my manuscript.

Best regards!

Ling Li

---

## [Decision Letter · Decision Letter 1]

8 Aug 2022

Framework for feature selection of predicting the diagnosis and prognosis of necrotizing enterocolitis

PONE-D-22-08369R1

Dear Dr. Li,

We’re pleased to inform you that your manuscript has been judged scientifically suitable for publication and will be formally accepted for publication once it meets all outstanding technical requirements.

Kind regards,

Vijayalakshmi Kakulapati, Ph.D

Academic Editor

PLOS ONE

---

## [Editor Report · Acceptance letter]

10 Aug 2022

PONE-D-22-08369R1 

Framework for feature selection of predicting the diagnosis and prognosis of necrotizing enterocolitis 

Dear Dr. Li:

I'm pleased to inform you that your manuscript has been deemed suitable for publication in PLOS ONE. Congratulations! Your manuscript is now with our production department. 

Kind regards, 

on behalf of

Dr. Vijayalakshmi Kakulapati 

Academic Editor

PLOS ONE